# The Antimicrobial Peptide D-CONGA-Q7 Eradicates Drug-Resistant *E. coli* by Disrupting Bacterial Cell Membranes

**DOI:** 10.3390/biology14030226

**Published:** 2025-02-21

**Authors:** Zonghan Jiang, Leisheng Sun, Yuanyuan Li, Haoyu Li, Yu Fu, Jiyun Li, Zhiliang Sun

**Affiliations:** 1Hunan Engineering Technology Research Center of Veterinary Drugs, Hunan Agricultural University, Changsha 410128, China; jiangzonghan0622@163.com (Z.J.); 18075717238@163.com (Y.L.); haoyulii@163.com (H.L.); fuyu728915@163.com (Y.F.); 2Key Laboratory of Study and Discovery of Small Targeted Molecules of Hunan Province, Department of Pharmacy, School of Medicine, Hunan Normal University, Changsha 410013, China; leishengsun@hunnu.edu.cn

**Keywords:** bacterial enteritis, *Escherichia coli*, *Mlac* gene, D-CONGA-Q7

## Abstract

*Escherichia coli* is a ubiquitous pathogen, with certain strains causing severe diarrhea and intestinal illnesses, which pose significant health risks to both children and animals. As antimicrobial resistance continues to rise, traditional antibiotics are increasingly ineffective, highlighting the urgent need for novel therapeutic strategies. In this study, we investigated the antimicrobial peptide D-CONGA-Q7, aiming to evaluate its efficacy and underlying mechanism against drug-resistant *E. coli* strains. Our results demonstrate that D-CONGA-Q7 effectively disrupts bacterial cell membranes, leading to bacterial death, and exhibits comparable activity against polymyxin- and meropenem-resistant strains. In vivo experiments revealed that D-CONGA-Q7 significantly alleviated intestinal inflammation, reduced inflammatory cytokine levels, and promoted the repair of damaged intestinal tissues. Furthermore, D-CONGA-Q7 displayed good stability. These findings suggest that D-CONGA-Q7 not only represents a promising new approach to combat drug-resistant bacterial infections but also holds potential for clinical application.

## 1. Introduction

*E. coli* is a pathogen commonly found in the environment. Although it exists as part of normal intestinal flora, *E. coli* can cause diarrhea and other extraintestinal diseases [1]. Diarrhea leads to hundreds of millions of illnesses globally each year, remaining a primary cause of mortality among children under 5 years old [2]. Diarrhea attributable to diarrhea-causing *Escherichia coli* (DEC) comprises 30–40% of all diarrhea cases in developing nations; while concentrated in resource-limited areas, infectious diarrhea cases remain significant in high-income countries as well [3,4,5]. MDRbacteria are strains that are simultaneously resistant to multiple classes of commonly used antimicrobial agents. The emergence of MDR strains, including those resistant to polymyxin and meropenem, has compromised last-line treatments such as meropenem and polymyxin in managing multidrug-resistant infections [6]. The urgent need for new antibiotics or alternative therapies is driven by antibiotic misuse, which has accelerated bacterial resistance at a rate surpassing the development of new antimicrobial agents. Developing new antibiotics has been a long-standing challenge [7,8].

The treatment of pathogenic bacterial infections has traditionally relied on antibiotics. However, the emergence of antibiotic resistance, driven by their single-target mechanisms, and prolonged [9] antimicrobial resistance and persistence are linked to a heightened risk of treatment failure and recurrent infections. Consequently, they are significant contributors to elevated morbidity, mortality, and healthcare costs, and the number of deaths due to drug resistance is increasing every year [10]. In contrast, antimicrobial peptides (AMPs) exhibit distinct advantages by targeting multiple sites on the plasma membrane and intracellular components of pathogenic bacteria, demonstrating potent efficacy against resistant bacterial strains [11,12], making them promising alternatives to traditional drugs against drug-resistant bacteria. SAAP-148, a novel AMP inspired by the human AMP LL-37, demonstrates broad-spectrum antimicrobial activity and low host cytotoxicity against MDR strains and has undergone clinical trials [13]. This example strongly demonstrates the feasibility of AMPs in killing MDR pathogens. However, AMPs face limitations such as uncontrolled in vivo toxicity, rapid metabolism, instability, and lack of oral bioavailability, which have hindered further research and clinical trials [9,14,15].

D-CONGA-Q7 (rrwarrqlafafrr-amide) is a D-type peptide synthesized by modifying D-CONGA with polar glutamate; it exhibits broad-spectrum in vitro antimicrobial activity and minimal erythrocyte hemolytic activity [16]. D-CONGA was originally screened from a library of 28,000 peptides using high-throughput synthesis principles and host cell compatibility [17]. More importantly, D-CONGA-Q7 has been shown to be bactericidal against drug-resistant Klebsiella pneumoniae, Pseudomonas aeruginosa, and Staphylococcus aureus in the previous period [16]. This study explored the antimicrobial mechanism of D-CONGA-Q7 through scanning electron microscopy (SEM), transmission electron microscope (TEM) and RNA sequencing. Additionally, in an *E. coli K88*-induced intestinal inflammation model, D-CONGA-Q7 demonstrated both potent antibacterial activity and anti-inflammatory effects. This study offers new strategies for combating drug-resistant bacteria and provides a theoretical foundation for developing D-CONGA-Q7 as a novel treatment.

## 2. Materials and Methods

### 2.1. Strains and Reagents

The bacterial strains used in this study, including *LN175*, *J27ab-2*, *K88*, *AD21R*, and *21f54mr-1*, were *E. coli* obtained from the Hunan Veterinary Engineering Research Center (Changsha, Hunan Province). Except for *K88*, all the strains were multidrug-resistant and carried multiple resistance genes, while *K88* was a diarrhea-causing strain containing the ETEC gene. D-CONGA-Q7 was purchased from GL Biochem (Shanghai) Ltd. (Shanghai, China).

### 2.2. Bacterial Culture

We retrieved the *E. coli* that had been stored at −80 °C, inoculated it onto LB (Land Bride, Beijing, China) solid media in a sterile workbench using streaking, and incubated it inverted at 37 °C for 16–24 h. A single colony was then selected and inoculated into 1 mL of LB liquid medium (Land Bride, Beijing, China), which was followed by incubation at 180 rpm at 37 °C overnight in a shaker. Subsequently, the culture was expanded at a 1:100 ratio to the desired concentration.

### 2.3. Animals and Treatment

Experimental animals were purchased from Hunan SJA Laboratory Animal Co. (Changsha, China). ICR mice were acclimated for one week in a controlled environment with a 12 h light/dark cycle, relative humidity of 60 ± 10%, and temperature maintained at 24 ± 2 °C. Thirty female SPF-grade ICR mice were fasted for 12 h and randomly divided into five groups (n = 6) based on body weight: a blank control (0.9% NaCl), a 4 mg/kg D-CONGA-Q7 group, a 2 mg/kg D-CONGA-Q7 group, a 2 mg/kg polymyxin E(PME) group, and a model group. A small intestinal inflammation model was induced by gavaging mice with a sodium bicarbonate suspension of 10^9^ CFU/mL *E. coli* (*K88*) once; the amount of bacterial liquid gavage is 10 mL/kg. And treatment was administered 24 h later for 7 consecutive days. At the end of the experiment, mice were anesthetized with tribromoethanol, and serum was collected via orbital blood sampling. All experiments and sample collection procedures were conducted in compliance with the Chinese Animal Welfare Guidelines and were approved by the Institutional Animal Care and Use Committee of Hunan Agricultural University.

### 2.4. Determination of Minimal Inhibitory Concentration (MIC) by Microbroth Dilution Method

The MIC values of D-CONGA-Q7 as well as other antibiotics against *E. coli* were determined by the microbroth dilution method according to the recommendations of the Clinical and Laboratory Standards Institute (CLSI), Malvern, PA, USA. The bacterial concentration was adjusted to 1 × 10^5^ CFU/mL using a turbidimeter. Then, 100 μL of Mueller–Hinton Broth (MHB) broth was added to wells 1–11 of a 96-well plate, while 200 μL was added to well 12. Then, 100 μL of the drug was added to wells 1–11, which was followed by 100 μL of the bacterial suspension. Results were observed after incubation at 37 °C for 16–24 h. MIC results were interpreted based on CLSI MD100-ED30 (2024) standards [18].

### 2.5. Determination of Minimum Bactericidal Concentration (MBC) of Peptides

To determine the minimum bactericidal concentration (MBC) of peptide, 100 μL of bacterial cultures was drawn from MIC and higher concentrations and spread evenly on MH agar plates. The plates were incubated at 37 °C for 48 h. Bacterial growth was assessed, and if fewer than five colonies appeared, the concentration was considered the MBC for that bacterium. Three replicates were set for each group.

### 2.6. D-CONGA-Q7 Stability Experiment

For thermal stability, the D-CONGA-Q7 was prepared as a 1.024 mg/mL solution and heated in a water bath at room temperature, 60, 65, 70, 75, 80, 90 and 100 °C for 1 h, and autoclaved at 121 °C for 15 min and then cooled to room temperature, respectively. Acid–base stability, 1.024 mg/mL of D-CONGA-Q7 solution was used to adjust its pH to 2–12 using HCl and NaOH, and the pH of each tube was adjusted back to the initial tube pH after 1 h. The adjustment of solution pH = 3.89 (pH = 3.89 is the pH of D-CONGA-Q7 itself when dissolved) included the above use of the untreated samples as a control.

The antimicrobial activity of D-CONGA-Q7 was evaluated by assessing changes in its MIC against strain LN175, K88, J27ab-2.

### 2.7. D-CONGA-Q7 Erythrocyte Hemolysis Assay

Peptide solutions at concentrations of 80 μg/mL, 40 μg/mL, and 20 μg/mL were prepared using 0.9% sodium chloride by serial dilution. To each test tube, 0.5 mL of a 2% sheep erythrocyte suspension (1001339-1) (Hopebio, Qingdao, China) was added, along with 0.5 mL of purified water, the solvent (0.9% NaCl), and the peptide test solution at varying concentrations. The tubes were mixed thoroughly and incubated at 37 °C for 3 h. The samples were then centrifuged at 2500 r/min for 5 min, and the absorbance of each tube was measured at 545 nm using a spectrophotometer.

### 2.8. Cell Culture and Activity Determination

Porcine small intestinal epithelial cells (IPEC-J2) were cultured in DMEM-F12 medium supplemented with 10% fetal bovine serum (TransGen Biotech, Beijing, China) and 1% penicillin–streptomycin (Beyotime Biotechnology, Shanghai, China). The cells were maintained in a humidified incubator at 37 °C with 5% CO_2_. Cells were passaged using 0.25% trypsin solution containing ethylenediaminetetraacetic acid (EDTA). Cells were seeded into 96-well plates at a density of 1 × 10⁴ cells per well. After 48 h, they were treated with varying concentrations of D-CONGA-Q7 and incubated with CCK-8 (Beyotime Biotechnology, Shanghai, China) for 1 h. The assay was measured using a fluorescence plate reader.

### 2.9. Determination of Bacterial Inner and Outer Membrane Permeability

Propidium iodide (PI) and N-phenyl-1-naphthylamine(NPN) were used to assess the integrity of the inner and outer membranes of *E. coli*. The bacterial suspension was diluted 1:100 in LB broth and incubated at 37 °C until reaching an OD_600nm_ of 0.6. The cells were then centrifuged, resuspended, and the OD_600nm_ was adjusted to 0.5 using PBS. NPN was added at a final concentration of 10 μM, and the mixture was incubated for 30 min at 37 °C in the dark. We added a final concentration of 0.25MIC 0.5MIC 1× MIC D-CONGA-Q7, incubated it for 1 h, and measured the fluorescence intensity using a fluorescence plate reader with excitation wavelengths of 535 nm (PI) and 350 nm (NPN) and emission wavelengths of 615 nm (PI) and 420 nm (NPN). Three biological replicates were prepared. NPN (P110559) and PI (P113815) were purchased from Aladdin Biochemical Technology Shanghai China.

### 2.10. ATP Assay

The bacterial ATP level was measured using an enhanced ATP assay kit (S0027) (Beyotime Biotechnology, Shanghai, China). *LN175* was cultured until OD_600nm_ = 0.5; then, 0.25MIC, 0.5MIC, and 1× MIC D-CONGA-Q7 were added, mixed, and incubated at 37 °C for 1 h. After incubation, the bacteria were washed three times with 1× PBS, centrifuged, and resuspended. The bacterial suspension was treated with 200 μL of 15 mg/mL lysozyme, vortexed, and the supernatant was collected. The sample and detection reagent were mixed in a 1:1 ratio in a black 96-well plate and measured using a luminometer. Three biological replicates were set for each group.

### 2.11. ROS Assay

The ROS level in bacteria was measured using a reactive oxygen species (ROS) detection kit (S0033M) (Beyotime Biotechnology, Shanghai, China). *LN175* was cultured until OD_600nm_ = 0.5; then, 0.25 MIC, 0.5 MIC, and 1× MIC D-CONGA-Q7 were added, mixed, and incubated for 1 h. After incubation, the 2′,7′-dichlorofluorescein diacetate (DCFH-DA) probe was loaded for 30 min with shaken every 3 min. After loading, the probe was washed three times with 1× PBS, and the ROS levels were measured using excitation at 488 nm and emission at 525 nm. Three biological replicates were performed for each group.

### 2.12. Transcriptome Processing, Scanning Electron Microscopy, and Transmission Electron Microscopy

Using a turbidimeter, the concentration of *E. coli LN175* was adjusted to 10^5^ CFU/mL and treatment with 1 μg/mL D-CONGA-Q7 for 4 h. Then, the bacteria were resuspended, washed twice with PBS, and centrifuged at 8000 rpm for 10 min. The supernatant was discarded, and the bacterial pellet was divided into two groups. One group was snap-frozen in liquid nitrogen for transcriptome sequencing, while the other group was fixed overnight at 4 °C with tissue fixative and analyzed using scanning electron microscopy (Hitachi, Tokyo, Japan, SU8100) and transmission electron microscopy (Hitachi, HT7800).

### 2.13. Histopathological Studies (HE)

Small intestine tissues were fixed in 4% paraformaldehyde, dehydrated through an ethanol gradient, and embedded in paraffin. Paraffin-embedded tissues were sectioned into 5 μm thick slices using a microtome and then mounted on silicone-coated slides for subsequent drying and deparaffinization. Sections were stained with hematoxylin and eosin (servicebio, G1076), which was followed by dehydration, permeabilization, and sealing. Finally, stained sections were observed under a light microscopy (Nikon Eclipse E100, Tokyo, Japan).

### 2.14. Enzyme-Linked Immunosorbent Assay (ELISA)

After blood collection, serum was separated by allowing the blood to clot, and the levels of IL-18, IL-6, and TNF-α in the serum of the mouse small intestine inflammation model were measured using ELISA kits (Jianglai biology, Shanghai, China), following the manufacturer’s instructions

### 2.15. Statistical Analysis

After conducting 3 independent replicates, the test data were graphed using GraphPad Prism 8.0.2 and statistically analyzed with SPSS 23.0. The results were presented as “mean ± standard error (X ± SEM)” with *p* < 0.05 indicating a significant difference and *p* < 0.01 indicating an extremely significant difference.

## 3. Result

### 3.1. D-CONGA-Q7 Exhibits Potent Activity Against Multidrug-Resistant E. coli Along with Good Stability and Low Cytotoxicity

We evaluated the MIC values of common antibiotics and D-CONGA-Q7 against a variety of multidrug-resistant *E. coli* strains as well as the MBC values of D-CONGA-Q7 for these bacteria. As shown in Figure 1A, D-CONGA-Q7 effectively eradicated multidrug-resistant *E. coli*, with MIC values less than or equal to 4 μg/mL for all strains tested, and we found that the MIC values are almost the same as the MBC values. From these, we selected *LN175*, which is resistant to most antibiotics and has multiple resistance genes, for follow-up. Notably, D-CONGA-Q7 showed high efficacy against polymyxin- and meropenem-resistant bacteria. Safety evaluation revealed no hemolysis at a concentration of 40 μg/mL with only a 1.7% hemolysis rate (Figure 1B). The IC50 value of D-CONGA-Q7 on IPEC-J2 porcine small intestinal epithelial cells was determined to be 188.5 μg/mL, indicating low cytotoxicity (Figure 1C). Furthermore, D-CONGA-Q7 retained its antimicrobial activity after exposure to temperatures ranging from 20 to 121 °C and pH levels between 2 and 11. A reduction in activity was only observed at pH 12, indicating that D-CONGA-Q7 is stable under high temperature and acidic conditions (Figure 1D,E). In conclusion, we demonstrated that D-CONGA-Q7 exhibits potent antimicrobial activity against multidrug-resistant *E. coli* with high stability and low cytotoxicity. 

### 3.2. D-CONGA-Q7 Exerts Its Bactericidal Effects by Altering the Permeability of the Inner and Outer Membranes and Disrupting the Integrity of the Bacterial Cell Wall

We first compared D-CONGA-Q7-treated *E. coli LN175* with the control group, finding that NPN and PI levels were elevated, while ATP levels decreased, and ROS levels increased; all parameters exhibited a positive correlation with D-CONGA-Q7 concentration (Figure 2A–D). This indicates that D-CONGA-Q7 rapidly enhances the permeability of both the inner and outer membranes, thereby disrupting the internal homeostatic state of the bacteria. To investigate whether D-CONGA-Q7 disrupts the bacterial cell wall and contributes to bacterial death, we examined the morphology and structure of D-CONGA-Q7-treated *E. coli LN175* using SEM. SEM observations revealed that D-CONGA-Q7-treated *E. coli LN175* exhibited crumpled and perforated cell walls compared to the control group, indicating that D-CONGA-Q7 disrupts the integrity of the bacterial cell wall. TEM further confirmed that the bacterial structure was damaged, the cell membrane was ruptured, and intracellular contents were exuded. In Appendix A, we counted the proportion of cell membrane crumpling under SEM and the proportion of cell membrane rupture under TEM. These results suggest that D-CONGA-Q7 can effectively eliminate bacteria by altering the permeability of the inner and outer membranes and disrupting the integrity of the bacterial cell membrane.

### 3.3. Transcriptome Analysis of LN175 Following D-CONGA-Q7 Treatment

Transcriptome analysis of LN175 after 4 h of D-CONGA-Q7 treatment, compared to the control group, revealed sixty-four genes with significant up-regulation and twenty-four genes with significant down-regulation (Appendix A, Figure 3). Based on previous experiments indicating that D-CONGA-Q7 primarily targets the bacterial cell membrane, we focused our analysis on genes associated with bacterial membrane proteins. Notably, we identified the genes *MlaC* and *YcfL*, with *MlaC* being a component of the *Mla* system, which is responsible for the bidirectional transport of phospholipids across the inner and outer membranes of Gram-negative bacteria, thereby maintaining bacterial homeostasis. The significant up-regulation of *MlaC* suggests that D-CONGA-Q7 may disrupt the phospholipid integrity of the inner and outer membranes, consequently impairing the homeostasis of these membranes and affecting the overall stability of the bacterial cell wall. Furthermore, the up-regulation of the gene *YcfL* indicates potential damage to bacterial DNA caused by reactive oxygen species (ROS).

### 3.4. Animal Experimentation

We observed that the mice exhibited no obvious symptoms when the gavage dose reached up to 500 mg/kg. This suggests that D-CONGA-Q7 exhibits lower toxicity when administered via gavage. Consequently, we evaluated the therapeutic effects of D-CONGA-Q7 on a bacterial enterocolitis model. Serum ELISA data showed that the medicated groups (4 mg/kg D-CONGA-Q7 and 2 mg/kg D-CONGA-Q7) had significantly reduced levels of inflammatory factors TNF-α, IL-6, and IL-18 compared to the model group. Some metrics in the low-dose D-CONGA-Q7 group were even superior to those in the polymyxin E treatment group (Figure 4E–G). Histological examination via HE staining revealed significant intestinal damage in the model group, including epithelial detachment at the villus tips, a marked reduction in villus length (*p* < 0.001), an increase in crypt depth (*p* < 0.001), and a decreased villus length/crypt depth ratio compared to the drug-treated groups. Furthermore, the intestinal muscularis propria was significantly thinned (Figure 4B–D), suggesting that D-CONGA-Q7 has great potential to treat bacterial enteritis.

## 4. Discussion

Since their discovery, antibiotics have been essential for treating bacterial infections, saving millions of lives annually. However, the global spread of MDR strains has rendered these infections increasingly difficult to treat [19,20].

AMPs, a class within the antibiotic family, typically carry cations and exhibit strong affinity for lipopolysaccharides in the outer membrane of Gram-negative bacteria, allowing them to quickly localize to bacterial cell membranes and exert antibacterial activity through membrane mediation [21,22]. However, due to the in vivo toxicity and instability of AMPs, only polymyxins B and E for Gram-negative bacteria and daptomycin for Gram-positive bacteria are FDA-approved for human use [23].

D-CONGA-Q7 is a D-type antimicrobial peptide initially screened through high-throughput peptide synthesis and host–cell compatibility principles and was subsequently structurally modified. Testing revealed that D-CONGA-Q7 can kill multidrug-resistant bacteria resistant to polymyxin, tigecycline, and meropenem, among others, while also exhibiting low hemolytic activity, low cytotoxicity, and strong acid–base and thermal stability. Furthermore, D-CONGA-Q7 was found to enhance the viability of *IPEC-J2* cells at a low concentration of 50 μg/mL.

*E. coli* has garnered significant attention since its discovery by Theodor Escherich in 1885. *E. coli* serves both as a commensal organism within the human body and as a pathogen causing diarrhea and extraintestinal diseases in humans and various animals [3]. Notably, enterotoxigenic *E. coli* (ETEC) and enteropathogenic *E. coli* (EPEC) harbor virulence factors encoded by mobile genetic elements, such as plasmids, which facilitate transfer between colonies, contributing to a mixed pathogenic phenotype [24]. ETEC produce heat-stable (ST) and heat-labile (LT) enterotoxins, which incite bacterial inflammation in the small intestine, leading to significant intestinal fluid accumulation and, ultimately, diarrhea [4,25]. Therefore, we selected *E. coli* strain *K88*, which possesses the ETEC virulence factor, as the causative agent of bacterial enteritis. Studies indicate that porcine β-defensin 2 exhibits a direct bactericidal effect and modulates the *TLR4/NF-κB* pathway, thereby inhibiting pro-inflammatory cytokines such as IL-6 [26,27]. Our findings demonstrate that D-CONGA-Q7 significantly reduces the levels of pro-inflammatory cytokines TNF-α, IL-6, and IL-18, effectively alleviating small intestinal inflammation in a mouse model of enteritis induced by *K88* (ETEC) exposure.

Subsequently, we assessed the permeability alterations of bacterial inner and outer membranes following D-CONGA-Q7 treatment, alongside transcriptomic changes, bacterial morphology, and membrane damage, utilizing SEM and TEM. The results indicated that D-CONGA-Q7 rapidly increased the permeability of both inner and outer membranes, resulting in significant perforation and content exudation from the bacteria, thereby suggesting that D-CONGA-Q7 functions as an antimicrobial peptide (AMP) capable of targeting the outer membrane (OM). Consequently, we concentrated on bacterial membranes to analyze the transcriptomic results and observed an up-regulation of the *MlaC* gene, which is involved in the lipid asymmetry *Mla* system responsible for phospholipid (PL) transport between the outer membrane (OM) and the inner membrane (IM). The *Mla* system is classified within the family of ATP-binding cassette (ABC) transport proteins, maintaining the outer membrane barrier via phospholipid (PL) transport between the IM and OM [28,29,30]. The OM of Gram-negative bacteria serves as a crucial barrier against antibiotics and environmental threats [28]. The OM is separated from the IM by the periplasmic space; consequently, double-membrane bacteria have evolved a series of transporter protein systems to indirectly obtain ATP, proton motive force, PL, and other substances from the cytoplasm. The up-regulation of the *Mlac* gene indicates that D-CONGA-Q7 may disrupt the outer membrane by affecting the PL composition, ultimately leading to bacterial death. Recent studies have identified *MlaC* as a potential therapeutic target for developing effective treatments against pathogenic Gram-negative bacteria. Concurrently, our analysis of the transcriptomic data revealed an elevation in the *YcfL* gene, which is associated with DNA damage resulting from reactive oxygen species (ROS) [31]. Studies have demonstrated that LL-37 induces both bacterial intracellular ROS production and uptake, and analogous results were observed with an increase in bacterial intracellular ROS following D-CONGA-Q7 treatment, suggesting an additional pathway through which D-CONGA-Q7 exerts its bactericidal effect [32,33].

While our study demonstrates that D-CONGA-Q7 effectively kills multidrug-resistant *E. coli* at low concentrations and exhibits therapeutic efficacy comparable to polymyxin E in a mouse model of small intestinal inflammation, certain limitations remain. For instance, although we demonstrated through SEM and TEM observations that D-CONGA-Q7 induces bacterial death by increasing the permeability of both inner and outer membranes, accompanied by a rise in ROS, OM perforation, and content exudation, we have only proposed a hypothesis regarding the specific target of action, which fails to offer a theoretical foundation for the development of novel antimicrobial peptides. Furthermore, although D-CONGA-Q7 can effectively treat mouse small intestine, its anti-inflammatory mechanism is unclear. These issues warrant further investigation in future studies, particularly in molecular biology and proteomics. Investigating the mechanisms of action and anti-inflammatory properties of D-CONGA-Q7 could pave the way for its development as a next-generation antimicrobial peptide against multidrug-resistant bacteria.

## 5. Conclusions

In conclusion, D-CONGA-Q7 is an antimicrobial peptide that eliminates bacteria by disrupting their cell membranes, potentially targeting PL and leading to membrane rupture. Additionally, it has demonstrated significant therapeutic effects against bacterial enteritis. This study further elucidates the potential targets of D-CONGA-Q7 against *E. coli* and establishes a theoretical foundation for its subsequent clinical application.

## Figures and Tables

**Figure 1 biology-14-00226-f001:**
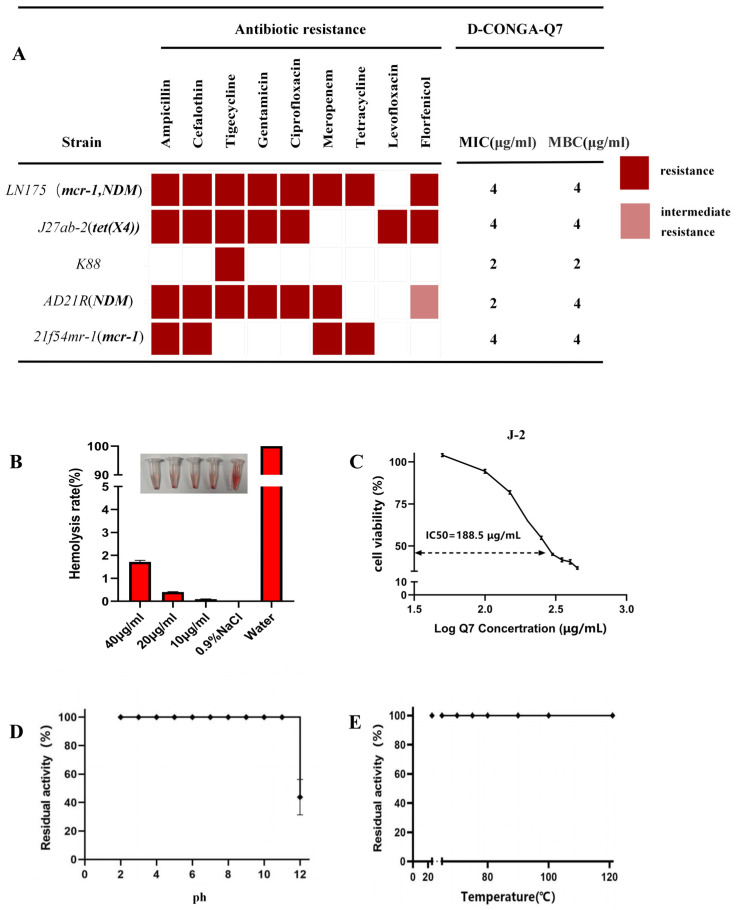
D-CONGA-Q7 exhibits bactericidal activity against multidrug-resistant *E. coli* along with high stability and low cytotoxicity. (**A**) MIC and MBC results based on CLSI 2024 standards. White boxes indicate sensitivity to antibiotics, red boxes indicate resistance, and pink boxes represent intermediate resistance, (**B**) Hemolysis assay, (**C**) Cytotoxicity assay, (**D**,**E**) Acid–base and thermal stability tests.

**Figure 2 biology-14-00226-f002:**
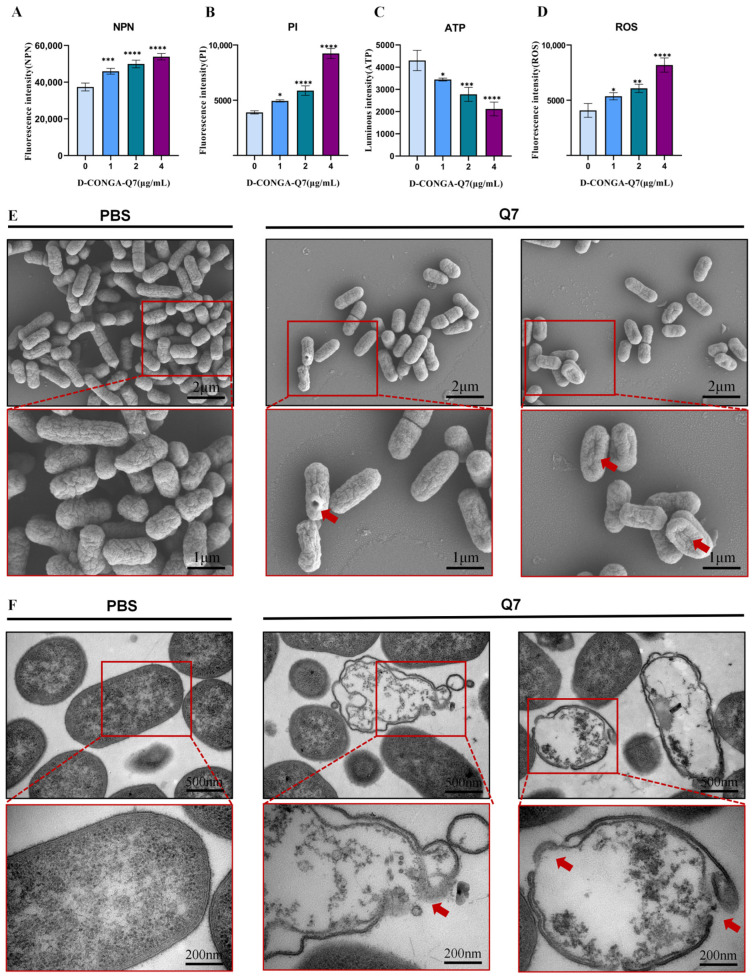
D-CONGA-Q7 exerts bactericidal effects by altering the permeability of the inner and outer membranes and compromising the integrity of the bacterial cell wall. (**A**–**D**) Changes in bacterial NPN, PI, ATP and ROS levels following 1 h of D-CONGA-Q7 treatment. (**E**) SEM observations of bacterial morphology after 4 h of D-CONGA-Q7 treatment with arrows indicating cell wall crumpling or perforation. (**F**) TEM observations of the bacterial cell wall after 4 h of D-CONGA-Q7 treatment with arrows indicating ruptured cell walls and exudation of cellular contents (scale bar = 1 μm). Data in panels (**A**–**D**) represent three biological replicates (n = 3), expressed as mean ± SD. Statistical significance compared to control is indicated as follows: * *p* < 0.05, ** *p* < 0.01, *** *p* < 0.001, **** *p* < 0.0001.

**Figure 3 biology-14-00226-f003:**
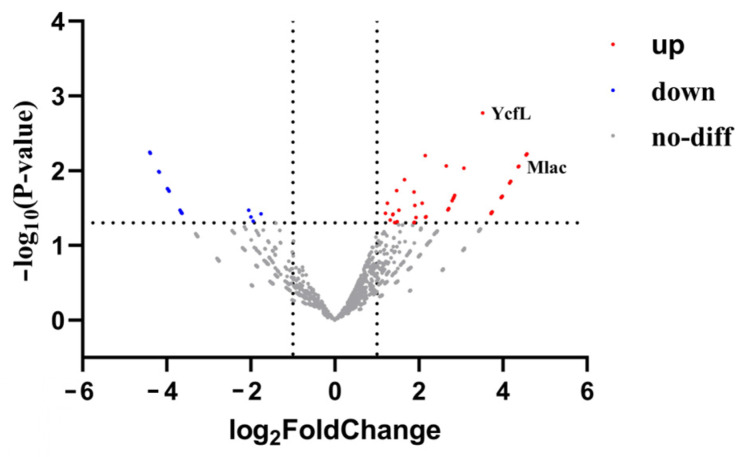
Transcriptome Analysis of *LN175* Following D-CONGA-Q7 Treatment. Red dots indicate significantly up-regulated genes, blue dots indicate significantly down-regulated genes, and gray dots indicate non-significantly different genes.

**Figure 4 biology-14-00226-f004:**
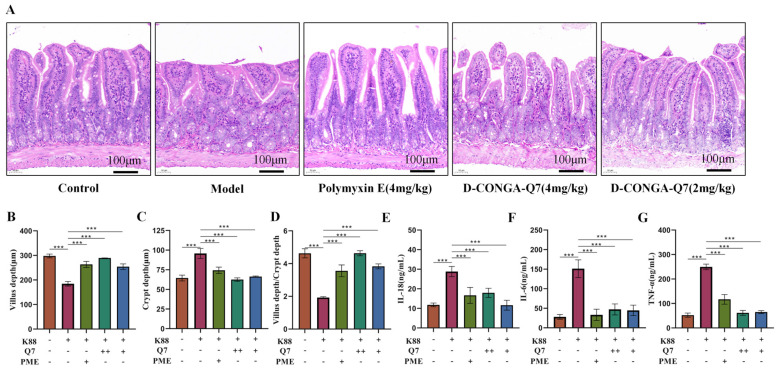
*K88*-induced intestinal inflammation model in D-CONGA-Q7-treated mice. (**A**–**D**) Small intestine H&E staining (scale bar = 100 μm), (**E**–**G**) ELISA detection of intestinal IL-6, IL-18 and TNF-α levels in mice “-” indicates no drug or bacterial solution, “++” represents the 4 mg/kg D-CONGA-Q7 group, and “+” denotes the addition of either the bacterial solution, 2 mg/kg D-CONGA-Q7, or 2 mg/kg polymyxin E. n = 3 Values are expressed as mean ± SD. Compared with the model group, *** *p* < 0.001.

## Data Availability

The original contributions presented in this study are included in the article/Appendix A. Further inquiries can be directed to the corresponding author(s).

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
