# Peer review of "The Antimicrobial Peptide D-CONGA-Q7 Eradicates Drug-Resistant E. coli by Disrupting Bacterial Cell Membranes"

_biology, 2025, doi:10.3390/biology14030226_

Round 1
Reviewer 1 Report (Previous Reviewer 4)
Comments and Suggestions for Authors
Dear authors,
greetings!
The manuscript was adjusted as recommended and, in my opinion, it can be published in the present form.
Author Response
Dear Reviewer 1 Thank you very much for your recognition!
Reviewer 2 Report (Previous Reviewer 2)
Comments and Suggestions for Authors
The manuscript by Jiang and Sun et al. presents the antibacterial properties of peptide Q7 against MDR E. coli. The authors have revised the manuscript following the first review. However, I still have some concerns and questions; therefore, in my review, I will also refer to their responses to my comments on the initial version of the manuscript.
1. In the initial publication, the terms D-CONGA Q7 and Q7 were used interchangeably, which led me to question whether these names are intended to be used synonymously. In their response, the authors changed the peptide name to Q7. D-CONAGA Q7 is a peptide derived from the modification of D-CONGA as described by Ghimire et al 2023. In their publication, this peptide is consistently referred to as D-CONGA Q7; therefore, I believe that shortening the name to Q7 in the current manuscript is incorrect and inappropriate. Since the authors are using an existing product, they should retain its original name.
2. It would be beneficial to provide a more detailed introduction to the D-CONGA Q7 peptide, including its structure and properties, especially since this peptide has already been partially characterized and its antibacterial activity against P. aeruginosa and K. pneumoniae has been evaluated.
3. Additionally, please revise the citations in this paragraph as follows:
· Lines 64-67 refer to Ghimire at al, 2023.
· Lines 66-67 refer to Starr et al. 2020
· Lines 68-69 refer again to Ghimire et al. 2023
I suggest reviewing this paragraph carefully and revising it to ensure it is fully informative and correctly referenced.
4. The introduction of lines 37-41 has led to a duplication of information. I suggest removing lines 34-37, which were previously lacking proper citation.
5. I would recommend describing the methods in the same order as they are presented in the 'Results' section. This approach enhances the organization and coherence of the manuscript, making it easier to analyze the results without difficulty in locating the corresponding methods used to obtain them.
6. In the first version of the manuscript, I pointed out that presenting the MIC values for peptide Q7 is not equivalent to determining its ability to eradicate bacteria. MIC determines the concentration that inhibits bacterial growth, but it is not synonymous with the Minimum Bactericidal Concentration (MBC). The absence of growth does not imply that the bacterial cells are dead; they may be in a dormant (persister) or VBNC state. In response, the authors changed the description of the results in both Figure 1 and its caption, replacing MIC with MBC. In my opinion, this introduced greater confusion into the description.
Firstly, at the beginning of paragraph 3.1 (line 204), it is written: "We evaluated MBC values of common antibiotics and Q7…". While replacing MIC with MBC may be appropriate for Q7, this is not the case for antibiotics. For most antibiotics, MIC does not equal MBC due to the presence of heteroresistant, persister, and VBNC bacteria. Additionally, the Materials and Methods section does not describe how the MIC and MBC were measured for antibiotics (only for Q7).
To compare the effect of Q7, I recommend keeping the MIC values in Figure 1A (as in the original version of the manuscript). Furthermore, a section should be added that describes, either in text, table, or figure format, that for Q7, MIC also corresponds to MBC. This would highlight the antibacterial advantages of Q7 over antibiotics.
It is worth noting that currently, changing MIC to MBC does not apply to all experiments (e.g. Fig 2), which creates significant confusion.
7. In the experiments described in section 3.2, it is necessary to compare the results with the negative control – dead cells. In this experiment, various concentrations of Q7 are used, including 1xMIC, which according to the authors is equal to MBC. A similar level of fluorescence for NPN and IP in both the negative control and 1xMIC would clearly indicate the bactericidal activity of Q7.
8. Since different incubation times in liquid medium are used for various concentrations of Q7 across different experiments in the publication, I suggest adding a graph illustrating bacterial survival (or mortality) curves (based on CFU) over time at given MIC values (e.g., for a selected strain). This would allow for a better understanding of the overall condition of the culture at the measurement point, such as for SEM and TEM analyses.
9. Still, there is an inconsistency in the graph legend for Figure 2A related to ATP measurement. The 'y' axis is currently labeled as "fluorescence intensity," whereas it should correctly indicate "luminescence intensity" or RLU value to reflect the appropriate measurement parameter.
10. In the description of the SEM and TEM experiment results, it is essential to include the frequency of cellular changes and membrane perforations after D1 treatment. Specific numerical values should be provided in the text to support the conclusions drawn (the corresponding graphs could be attached in the supplementary materials).
11. The phrase "scale bar = 1 µm" should be removed from the description of Figure 2.
12. There is still a lack of consistency in the reported measurable quantities. In Section 2.3, the gavage dose was initially provided in CFU/kg body weight of the rat (in the first version of the manuscript); however, it has now been changed to ml/kg, whereas in the Results section (line 271), the gavage dose is mentioned in mg/kg. This should be revised.
13. I suggest clarifying the meaning of the abbreviation "PME" (line 276), which is also used in Figure 4.
14. In Figure 4A, the scale should be clearly indicated, similar to how it was done in Figure 2. (and also the phrase "scale bar = 1 µm" should be removed from the description of Figure 4)
15. Style and Nomenclature:
- The names of bacterial species should be written in italics. Please ensure that when a species is first mentioned in the manuscript, its full name is used (Escherichia coli). Subsequent references to the species should be abbreviated (E. coli). This should be corrected throughout the entire manuscript.
- Gene names should be written in lowercase and italics (e.g., mlaC, https://biocyc.org/gene?orgid=ECOLI&id=G7659). This should be corrected throughout the entire manuscript.
- Please follow the correct citation format according to the author guidelines (References must be numbered in order of appearance in the text) – line 292
- The incorrect keyword "bacillary enteritis" should be corrected to "bacterial enteritis."
- Please standardize the style throughout the manuscript and use lowercase for measurement units (e.g., mg/ml, not mg/mL).
- Please pay attention to the correctness of the style, especially in the inserted fragments (periods at the end of sentences, capital letters at the beginning of sentences, etc.)
Author Response
Detailed reply to reviewer’s comments:
To Reviewer
Dear Reviewer, Thank you very much for your valuable comments. Your Suggestions have helped us to improve the deficiencies in the manuscript and increase the rigor and readability of the article. We have completed the revision and have responded to your comments point by point as follows:
Majors:
Comment 1. [ In the initial publication, the terms D-CONGA Q7 and Q7 were used interchangeably, which led me to question whether these names are intended to be used synonymously. In their response, the authors changed the peptide name to Q7. D-CONAGA Q7 is a peptide derived from the modification of D-CONGA as described by Ghimire et al 2023. In their publication, this peptide is consistently referred to as D-CONGA Q7; therefore, I believe that shortening the name to Q7 in the current manuscript is incorrect and inappropriate. Since the authors are using an existing product, they should retain its original name.]
Response 1.[Dear reviewers, thank you for your careful review and constructive suggestions regarding our manuscript. In response to your suggestions, we have revised the manuscript. We have changed the name Q7 in the manuscript to D-CONGA-Q7.]
Comment 2. [It would be beneficial to provide a more detailed introduction to the D-CONGA Q7 peptide, including its structure and properties, especially since this peptide has already been partially characterized and its antibacterial activity against P. aeruginosa and K. pneumoniae has been evaluated.]
Response 2.[Thank you for your valuable feedback on our paper. based on your valuable comments.We have added the results of inhibition experiments on Pseudomonas aeruginosa and Klebsiella pneumoniae from previous articles on D-CONGA-Q7.]
Comment 3. [Additionally, please revise the citations in this paragraph as follows:
- Lines 64-67 refer to Ghimire at al, 2023.
- Lines 66-67 refer to Starr et al. 2020
- Lines 68-69 refer again to Ghimire et al. 2023
I suggest reviewing this paragraph carefully and revising it to ensure it is fully informative and correctly referenced.]
Response 3.[Thank you very much for your kind attention. We have corrected the quotation in the manuscript in the appropriate place]
Comment 4. [The introduction of lines 37-41 has led to a duplication of information. I suggest removing lines 34-37, which were previously lacking proper citation.]
Response 4.[Thank you for your thorough review and we apologize for our carelessness. We have removed duplicate information from the manuscript]
Comment 5. [ I would recommend describing the methods in the same order as they are presented in the 'Results' section. This approach enhances the organization and coherence of the manuscript, making it easier to analyze the results without difficulty in locating the corresponding methods used to obtain them.]
Response 5.[Thank you for your thorough review and we apologize for our carelessness.We have aligned the order of material methods in the manuscript with those in the results]
Comment 6. [ In the first version of the manuscript, I pointed out that presenting the MIC values for peptide Q7 is not equivalent to determining its ability to eradicate bacteria. MIC determines the concentration that inhibits bacterial growth, but it is not synonymous with the Minimum Bactericidal Concentration (MBC). The absence of growth does not imply that the bacterial cells are dead; they may be in a dormant (persister) or VBNC state. In response, the authors changed the description of the results in both Figure 1 and its caption, replacing MIC with MBC. In my opinion, this introduced greater confusion into the description.]
Firstly, at the beginning of paragraph 3.1 (line 204), it is written: "We evaluated MBC values of common antibiotics and Q7…". While replacing MIC with MBC may be appropriate for Q7, this is not the case for antibiotics. For most antibiotics, MIC does not equal MBC due to the presence of heteroresistant, persister, and VBNC bacteria. Additionally, the Materials and Methods section does not describe how the MIC and MBC were measured for antibiotics (only for Q7).
To compare the effect of Q7, I recommend keeping the MIC values in Figure 1A (as in the original version of the manuscript). Furthermore, a section should be added that describes, either in text, table, or figure format, that for Q7, MIC also corresponds to MBC. This would highlight the antibacterial advantages of Q7 over antibiotics.
It is worth noting that currently, changing MIC to MBC does not apply to all experiments (e.g. Fig 2), which creates significant confusion.]
Response 6.[Thank you very much for your helpful guidance. We placed the MIC and MBC results in Figure 1 and enhanced the subsequent analysis to show that D-CONGA-Q7's MIC is consistent with MBC for some bacteria]
Comment 7. [ In the experiments described in section 3.2, it is necessary to compare the results with the negative control – dead cells. In this experiment, various concentrations of Q7 are used, including 1xMIC, which according to the authors is equal to MBC. A similar level of fluorescence for NPN and IP in both the negative control and 1xMIC would clearly indicate the bactericidal activity of Q7.]
Response 7.[Thank you for your careful inspection.Although the Q7 concentration used in the description of the results in 3.2 is MBC, the treatment time using MBC concentration Q7 is only 1h, which is not enough to kill all the bacteria, and NPN and PI are detecting the change of the permeability of the bacterial membrane, and at the 1h detection, the bacteria should be in a poor state but not completely dead, so we chose 1h as the time to measure the permeability.]
Comment 8. [ Since different incubation times in liquid medium are used for various concentrations of Q7 across different experiments in the publication, I suggest adding a graph illustrating bacterial survival (or mortality) curves (based on CFU) over time at given MIC values (e.g., for a selected strain). This would allow for a better understanding of the overall condition of the culture at the measurement point, such as for SEM and TEM analyses.]
Response 8.[Thank you very much for your valuable comments.We did about the killing effect of Q7 on LN175 at different concentrations at different times, we didn't put the data and results in this article, not that this data is not important for this article but for the sake of completeness of the next article.We apologize for this.]
Comment 9. [Still, there is an inconsistency in the graph legend for Figure 2A related to ATP measurement. The 'y' axis is currently labeled as "fluorescence intensity," whereas it should correctly indicate "luminescence intensity" or RLU value to reflect the appropriate measurement parameter.]
Response 9.[Thank you for your thorough review and we apologize for our carelessness.The corresponding Y-axis description has been changed]
Comment 10. [In the description of the SEM and TEM experiment results, it is essential to include the frequency of cellular changes and membrane perforations after D1 treatment. Specific numerical values should be provided in the text to support the conclusions drawn (the corresponding graphs could be attached in the supplementary materials).]
Response 10.[Thank you very much for your valuable comments,In the Supplementary Material, information has been added on the frequency of SEM and TEM perforations, as well as the specific number of]
Comment 11.[ The phrase "scale bar = 1 µm" should be removed from the description of Figure 2.]
Response 11.[Thank you for your thorough review and we apologize for our carelessness.Changes have been made in a new manuscript.]
Comment 12.[There is still a lack of consistency in the reported measurable quantities. In Section 2.3, the gavage dose was initially provided in CFU/kg body weight of the rat (in the first version of the manuscript); however, it has now been changed to ml/kg, whereas in the Results section (line 271), the gavage dose is mentioned in mg/kg. This should be revised.]
Response 12.[Thank you very much for your valuable comments.I'm sorry I didn't understand what you meant, but the mg/kg in the results all refer to the drug concentration.]
Comment 13.[ I suggest clarifying the meaning of the abbreviation "PME" (line 276), which is also used in Figure 4.]
Response 13.[Thank you very much for pointing it out.The PME abbreviation has been changed to the full name in a new manuscript]
Comment 14.[ In Figure 4A, the scale should be clearly indicated, similar to how it was done in Figure 2. (and also the phrase "scale bar = 1 µm" should be removed from the description of Figure 4)]
Response 14.[Thank you very much for your helpful guidance.Scale of figure 4 has been updated in a new manuscript
Comment 15.[ Style and Nomenclature:
- The names of bacterial species should be written in italics. Please ensure that when a species is first mentioned in the manuscript, its full name is used (Escherichia coli). Subsequent references to the species should be abbreviated (E. coli). This should be corrected throughout the entire manuscript.
- Gene names should be written in lowercase and italics (e.g., mlaC, https://biocyc.org/gene?orgid=ECOLI&id=G7659). This should be corrected throughout the entire manuscript.
- Please follow the correct citation format according to the author guidelines (References must be numbered in order of appearance in the text) – line 292
- The incorrect keyword "bacillary enteritis" should be corrected to "bacterial enteritis."
- Please standardize the style throughout the manuscript and use lowercase for measurement units (e.g., mg/ml, not mg/mL).
- Please pay attention to the correctness of the style, especially in the inserted fragments (periods at the end of sentences, capital letters at the beginning of sentences, etc.)]
Response 15.[Thank you very much for your valuable comments.The strain name and gene name have been italicized in the manuscript, the citation format has been changed, the keywords have been corrected to bactrial enteritis, and the unit of measurement has been standardized]
Finally, we appreciate for academic editor and reviewers’ warm work and patience earnestly.
Reviewer 3 Report (Previous Reviewer 1)
Comments and Suggestions for Authors
The revised version has been improved significantly.
Author Response
Dear Reviewer 3 Thank you very much for your recognition!
Reviewer 4 Report (Previous Reviewer 3)
Comments and Suggestions for Authors
The authors state in the abstract: "In conclusion, this study provides a novel approach to combat drug-resistant E. coli and establishes a theoretical basis for the design of more effective antibiotics. However, it has been known for decades that AMPs disrupt bacterial membranes. However, this knowledge lacks a structure-function relationship with respect to the molecular mechanisms of AMP-lipid interactions. In this respect, the present study does not provide a theoretical basis for the development of new AMPs, but only shows the effect of the Q7 peptide on E. coli. This should be clearly stated in the abstract and conclusions.
The manuscript has many spelling mistakes like ph instead of pH, hitachi instead of Hitachi, and so on. Please read it carefully and correct all the problems.
Author Response
Detailed reply to reviewer’s comments:
To Reviewer
Dear Reviewer, Thank you very much for your valuable comments. Your Suggestions have helped us to improve the deficiencies in the manuscript and increase the rigor and readability of the article. We have completed the revision and have responded to your comments point by point as follows:
Majors:
Comment 1. [ The authors state in the abstract: "In conclusion, this study provides a novel approach to combat drug-resistant E. coli and establishes a theoretical basis for the design of more effective antibiotics. However, it has been known for decades that AMPs disrupt bacterial membranes. However, this knowledge lacks a structure-function relationship with respect to the molecular mechanisms of AMP-lipid interactions. In this respect, the present study does not provide a theoretical basis for the development of new AMPs, but only shows the effect of the Q7 peptide on E. coli. This should be clearly stated in the abstract and conclusions.
The manuscript has many spelling mistakes like ph instead of pH, hitachi instead of Hitachi, and so on. Please read it carefully and correct all the problems.]
Response 1.[Dear reviewers, thank you for your careful review and constructive suggestions regarding our manuscript. In response to your suggestions, we have revised the manuscript. We have emphasized in our results that this experiment did not reveal a specific mechanism of action of AMP on bacteria, and could not provide a new theoretical basis for the subsequent development of novel antimicrobial peptides, and the description of ph and hitachi has been corrected in the new manuscript.]
Round 2
Reviewer 2 Report (Previous Reviewer 2)
Comments and Suggestions for Authors
The manuscript by Jiang and Sun et al. presents the antibacterial properties of the peptide D-CONGA-Q7 against MDR K. pneumoniae. The authors have revised the manuscript following the first review of the second version.
However, I still have a few minor comments regarding the manuscript:
- In Section 2.4, the authors have modified the wording to indicate that MBC was determined for both "peptides" and "other antibiotics." This paragraph requires further clarification. Based on the description of the results (lines 209–211) and the authors' previous explanations, MIC was determined for both the peptide and antibiotics, whereas MBC was tested only for the peptide. If MBC was also determined for antibiotics, these data should be included in the results section. Given the nature of antibiotics and their potential to induce a persister state or VBNC (Viable But Non-Culturable) state, it is likely that the MBC for antibiotics is higher than the MIC. In contrast, for the peptide, MIC equals MBC. Therefore, it would be beneficial to compare the MBC values of the peptide and antibiotics, rather than just MIC. If the manuscript focuses solely on MIC (noting that for the peptide, MIC equals MBC), the information in Section 2.4 regarding MBC determination for antibiotics should be removed. Additionally, this section refers to "peptides" (plural), whereas only one peptide (D-CONGA-Q7) was tested. This should be corrected in both the section title and the text of Section 2.4.
- In Section 2.1, it would be appropriate to include the species name of the bacteria, as currently, only strain names are listed.
- In my previous review, I suggested: "It would be beneficial to provide a more detailed introduction to the D-CONGA-Q7 peptide, including its structure and properties, especially since this peptide has already been partially characterized and its antibacterial activity against P. aeruginosa and K. pneumoniae has been evaluated." In their response, the authors stated: "We have added the results of inhibition experiments on Pseudomonas aeruginosa and Klebsiella pneumoniae from previous articles on D10." However, no such information has been added to the manuscript.
- There is still an error in citation placement. In line 65, the information corresponds only to [18], whereas the content of [17] is referenced in lines 65–67.
The manuscript contains numerous editorial inconsistencies. I have previously pointed these out in my earlier reviews, and despite the authors' response indicating that corrections have been made, these revisions are either not visible, only partially applied throughout the manuscript, or introduced incorrectly. Particular attention should be paid to these issues during the final editing of the text:
- Species names should be written in italics, both in full and abbreviated forms. The full name should be used upon first mention (lines 12 and 34), while the abbreviated form should be used thereafter (line 38 should include the abbreviated name). In the current manuscript, the abbreviated version lacks a space between the first letter of the genus name, followed by a period, and the species name.
- Strain names (e.g., K88) should not be italicized.
- Gene names should be written in lowercase italics. This has been corrected in some instances, but inconsistencies remain in lines 269, 275, 352, and 356.
- The manuscript contains multiple stylistic errors, such as:
- Sentences ending with a comma or incorrect capitalization after a comma, making it unclear whether these are separate sentences or a single sentence incorrectly divided by capitalization (e.g., lines: 50, 52, 56, 64, 96, 97, 125, 126, 211, 213, 215).
- Line 230 – Missing commas before figure letter labels.
- Line 299 – Missing parentheses before "E-G".
- Lines 369–370 – The sentence requires revision.
- It is unclear why the notation for pH has been changed to ph, as the standard scientific notation is pH.
- The notation for OD600 should be consistent throughout the manuscript (lines 15, 151, and 162).
- Concentrations are typically written as 1×MIC rather than 1MIC (lines 163, 172).
- In several places, spaces are missing before citation markers (e.g., lines 311, 313) or before parentheses (e.g., lines 185, 186, 192, 193).
- Lines 153–154 – The sentence is written in the present tense, while the rest of the paragraph is in the past tense.
Strengths: The study presents well-designed experiments that effectively and comprehensively demonstrate the advantages of the novel antibacterial agent D-CONGA-Q7. Additionally, it highlights the necessity for further research in this area.
Weaknesses: Numerous editorial errors, which reduce the overall readability and quality of the manuscript.
Author Response
Comments 1:[ in Section 2.4, the authors have modified the wording to indicate that MBC was determined for both "peptides" and "other antibiotics." This paragraph requires further clarification. Based on the description of the results (lines 209–211) and the authors' previous explanations, MIC was determined for both the peptide and antibiotics, whereas MBC was tested only for the peptide. If MBC was also determined for antibiotics, these data should be included in the results section. Given the nature of antibiotics and their potential to induce a persister state or VBNC (Viable But Non-Culturable) state, it is likely that the MBC for antibiotics is higher than the MIC. In contrast, for the peptide, MIC equals MBC. Therefore, it would be beneficial to compare the MBC values of the peptide and antibiotics, rather than just MIC. If the manuscript focuses solely on MIC (noting that for the peptide, MIC equals MBC), the information in Section 2.4 regarding MBC determination for antibiotics should be removed. Additionally, this section refers to "peptides" (plural), whereas only one peptide (D-CONGA-Q7) was tested. This should be corrected in both the section title and the text of Section 2.4.]
Response 1:[Dear reviewers, thank you for your careful review and constructive suggestions regarding our manuscript. In response to your suggestions,removed 2.4 information related to measuring MBC of other antibiotics and changed peptide name to D-CONGA-Q7.]
Comments 2:[In Section 2.1, it would be appropriate to include the species name of the bacteria, as currently, only strain names are listed.]
Response 2:[Thank you for your valuable feedback on our paper.We have labeled species names in 2.1.]
Comments 3:[In my previous review, I suggested: "It would be beneficial to provide a more detailed introduction to the D-CONGA-Q7 peptide, including its structure and properties, especially since this peptide has already been partially characterized and its antibacterial activity against P. aeruginosa and K. pneumoniae has been evaluated." In their response, the authors stated: "We have added the results of inhibition experiments on Pseudomonas aeruginosa and Klebsiella pneumoniae from previous articles on D10." However, no such information has been added to the manuscript.]
Response 3:[Thank you very much for your kind attention.In the new manuscript, the previous bactericidal effect against Pseudomonas aeruginosa as well as Klebsiella pneumoniae has been emphasized in the INTRODUCTION]
- Comments 4:[There is still an error in citation placement. In line 65, the information corresponds only to [18], whereas the content of [17] is referenced in lines 65–]
Response 4:[Thank you very much for your helpful comments.Have updated the introductory position in the new manuscript to be more accurate]
Comments 5:[The manuscript contains numerous editorial inconsistencies. I have previously pointed these out in my earlier reviews, and despite the authors' response indicating that corrections have been made, these revisions are either not visible, only partially applied throughout the manuscript, or introduced incorrectly. Particular attention should be paid to these issues during the final editing of the text,Species names should be written in italics, both in full and abbreviated forms. The full name should be used upon first mention (lines 12 and 34), while the abbreviated form should be used thereafter (line 38 should include the abbreviated name). In the current manuscript, the abbreviated version lacks a space between the first letter of the genus name, followed by a period, and the species name.]
Response 5:[Thank you for your thorough review and we apologize for our carelessness.In the new manuscript, the correct writing about strains has been corrected]
Comments 6:[Strain names (e.g., K88) should not be italicized.]
Response 6:[Thank you for your thorough review and we apologize for our carelessness.In the new manuscript, a correction has been made regarding the correct writing of the strain name]
Comments 7:[Gene names should be written in lowercase italics. This has been corrected in some instances, but inconsistencies remain in lines 269, 275, 352, and 356.]
Response 7:[Thank you for your valuable feedback on our paper.We have changed to italicize all gene abbreviations in the new manuscript change]
Comments 8:[The manuscript contains multiple stylistic errors, such as:
Sentences ending with a comma or incorrect capitalization after a comma, making it unclear whether these are separate sentences or a single sentence incorrectly divided by capitalization (e.g., lines: 50, 52, 56, 64, 96, 97, 125, 126, 211, 213, 215).]
Response 8:[Thank you very much for your kind attention.In the new manuscript, we checked for case as well as space issues]
Comments 9:[Line 230 – Missing commas before figure letter labels.Line 299 – Missing parentheses before "E-G".Lines 369–370 – The sentence requires revision.
It is unclear why the notation for pH has been changed to ph, as the standard scientific notation is pH.The notation for OD600 should be consistent throughout the manuscript (lines 15, 151, and 162).Concentrations are typically written as 1×MIC rather than 1MIC (lines 163, 172).In several places, spaces are missing before citation markers (e.g., lines 311, 313) or before parentheses (e.g., lines 185, 186, 192, 193).
Lines 153–154 – The sentence is written in the present tense, while the rest of the paragraph is in the past tense.]
Response 9:[Thank you very much for your kind attention.In the new manuscript, we have highlighted the writing problem several times and have revised it]
Reviewer 4 Report (Previous Reviewer 3)
Comments and Suggestions for Authors
The Abstract still contains a statement about theoretical basis for developing more effective antibiotics. Please, remove.
Author Response
Comments 1:[The Abstract still contains a statement about theoretical basis for developing more effective antibiotics. Please, remove.]
Response 1:[Dear reviewers, thank you for your careful review and constructive suggestions regarding our manuscript. In response to your suggestions,The corresponding part of the summary has been removed]
This manuscript is a resubmission of an earlier submission. The following is a list of the peer review reports and author responses from that submission.
Round 1
Reviewer 1 Report
Comments and Suggestions for Authors
The authors investigated the antimicrobial mechanism of an antimicrobial peptide (Q7) using scanning electron microscopy (SEM), transmission electron microscopy (TEM), and RNA sequencing.
The methodology is robust and effectively supports the study's conclusions.
I have a few minor comments and suggestions to further refine and strengthen the manuscript.
1) K88 is not a multidrug-resistant strain; It is included in this study due to its virulence (i.e., it leads to intestinal inflammation). Please ensure this distinction is clearly stated and consistently emphasized throughout the text.
2) Figure 1: Please define the abbreviations for the antibiotics listed in the figure for clarity.
3) Performing WGS could provide a deeper understanding of the resistome, virulome, and strain type of the tested isolates, enhancing the scope of the findings.
4) Could the antimicrobial peptide (Q7) inhibit specific virulence factors? Exploring this aspect could add valuable insights into its mechanism of action.
5) Can the tested antimicrobial peptide work synergistically with other antibiotics or restore the activity of antibiotics that are initially inactive against MDR E. coli strains?
6) In addition to discussing the limitations, please include the strengths of this study at the end of the discussion section to provide a balanced perspective.
7) Have you done a chemical analysis of Q7? This may be crucial.
Comments on the Quality of English LanguageNA
Author Response
To Reviewer #1:
Dear Reviewer, Thank you very much for your valuable comments. Your Suggestions have helped us to improve the deficiencies in the manuscript and increase the rigor and readability of the article. We have completed the revision and have responded to your comments point by point as follows:
Majors:
Comments 1:[ K88 is not a multidrug-resistant strain; It is included in this study due to its virulence (i.e., it leads to intestinal inflammation). Please ensure this distinction is clearly stated and consistently emphasized throughout the text.]
Response 1:[Dear reviewers, thank you for your careful review and constructive suggestions regarding our manuscript. In response to your suggestions,We have emphasized the pathogenicity and hazards of the strains in the revised materials section and the updated discussion.
Comments 2:[Figure 1: Please define the abbreviations for the antibiotics listed in the figure for clarity.]
Response 2:[Thank you for your valuable feedback on our paper.We have updated the full name of the antibiotic in Figure 1 and its legend to enhance clarity and improve comprehension.]
Comments 3:[Performing WGS could provide a deeper understanding of the resistome, virulome, and strain type of the tested isolates, enhancing the scope of the findings.]
Response 3:[Thank you very much for your kind attention.We have sequenced the whole genome of the isolates and found multiple resistance genes, which is why we chose these multi-drug resistant strains for our experiments, and have emphasized their resistance genes as well as multi-drug resistance in our new material approach.]
Comments 4:[Could the antimicrobial peptide (Q7) inhibit specific virulence factors? Exploring this aspect could add valuable insights into its mechanism of action.]
Response 4:[Thank you very much for your helpful guidance. Our team has recently explored whether Q7 can inhibit the virulence genes of E. coli. However, neither transcriptomic data nor epistasis analyses have revealed such inhibition. We speculate that Q7 may directly inhibit or kill bacteria in the intestinal tract, thereby alleviating small intestine inflammation.]
Comments 5:[Can the tested antimicrobial peptide work synergistically with other antibiotics or restore the activity of antibiotics that are initially inactive against MDR E. coli strains?]
Response 5:[Thank you for your kind comments. Since Q7 exhibited an MIC below 4 μg/ml against nearly all drug-resistant and sensitive bacteria, and its mechanisms of action and anti-inflammatory effects remain unclear, we have not pursued drug combination studies thus far. However, your suggestion has provided a valuable new research direction for future experiments.]
Comments 6:[In addition to discussing the limitations, please include the strengths of this study at the end of the discussion section to provide a balanced perspective.]
Response 6:[Thank you very much for your valuable comments,The benefits of Q7 have already been emphasized in the revised discussion section.]
Comments 7:[Have you done a chemical analysis of Q7? This may be crucial.]
Response 7:[We conducted liquid chromatography on the synthesized Q7, demonstrating that its purity was at least 99%.]
Reviewer 2 Report
Comments and Suggestions for Authors
The manuscript written by Jiang, Sun and colleagues aims to evaluate the effectiveness and action of the peptide against MDR E. coli and suggests its mode of action. Unfortunately, the manuscript is difficult to read due to an underdeveloped description of the methodology, sparse presentation of results, and poorly or insufficiently labeled figures. Although the experiments were well designed, their results are sparsely described and could benefit from a more detailed description. Additionally, the figures and their captions lack clear and informative legends. While the manuscript presents several valuable experiments, the overall quality is poor, and the work would benefit significantly from language editing (as well as the use of correct microbiological and molecular nomenclature). Below, I present my detailed suggestions and questions:
1. I suggest reworking the 'Introduction' section.
- Valuable information regarding E. coli, currently found in the 'Discussion' section (lines 277-293), should be moved to the introduction to better introduce the topic of pathogenicity.
- I recommend expanding the introductory information on AMPs (lines 45-46), e.g., by providing a definition of AMPs, describing their general common mechanisms of action, and highlighting their broad range of activity. It would also be valuable to mention that the mechanism of action of peptides makes them active against both metabolically active and dormant bacteria (including persisters and VBNC) in contrast to conventional antibiotics, which target specific molecular mechanisms requiring bacterial activity and do not act on dormant bacteria (e.g. 10.3390/antibiotics12061044 or others).
- I suggest removing lines 53-55. The content of this part is not coherent with the rest of the paragraph. It could be integrated into the paragraph discussing the modification introduced to D-CONGA (to explain why this modification was made).
- Please revise lines 57-63, as it is unclear, especially lines 57-58. Two peptides are described, but one name is used, which is confusing. As I understand, D-CONGA was initially selected (Starr et al, 2020), and then it was modified by Ghimire et al.,(2023), and named D-CONGA-Q7.
- Is the use of the name Q7 in the manuscript simply an abbreviation for D-CONGA-Q7? This should be clarified.
- Lines 31-33: Lack of references.
- Lines 33-34: Lack of references.
2. The 'Materials and Methods' section should be revised and expanded:
- I would recommend describing the methods in the same order as they are presented in the 'Results' section.
- Information about NPN and PI should be moved from subsection 2.1 to subsection 2.11.
- I would suggest expanding the information on the strains used in the study (a description of these strains or relevant references), or explaining why these specific strains were chosen for the experiments. This is particularly important for the selection of the K88 strain for mice infections (the information in lines 293-294 should be moved to the beginning of the 'Results' section or included in the description of strains in the 'Materials and Methods' section) and LN175 for other experiments.
- There is no information on how the MIC of the antibiotics was measured, even though line 18 states: "we evaluated the MIC values of common antibiotics...".
- There is no information on which strain was tested with PI and NPN or what concentrations of Q7 were added to the samples in subsection 2.11. This becomes clear only in figure 2A.
- There is no description of how the small intestine tissue samples were collected, fixed, and stained, nor how the microscopic observations were conducted.
- Subsection 2.4 does not specify which antibodies were used (company, origin).
- There is no mention of the specific equipment used for the experiments. For example, while a scanning electron microscope is mentioned, its exact model and the name or parameters of the camera used to capture the images are not provided.
3. The manuscript presents MIC results for Q7 and other antibiotics. MIC determines the concentration that inhibits bacterial growth, but it is not synonymous with the Minimum Bactericidal Concentration (MBC). The absence of growth does not imply that the bacterial cells are dead; they may be in a dormant state (persisters) or in a VBNC state. In the Materials and Methods section, it is stated that MBC was evaluated (subsection 2.6), but these results are not presented or analyzed. In the 'Results' section, there is no mention of MBC, but it is stated that "Q7 effectively eradicated multidrug-resistant E. coli with MIC values below 4 µg/ml" (lines 186-187). This statement is also mentioned in the manuscript title. MIC does not equate to bacterial eradication. MBC results should be provided, demonstrating that Q7 does not induce the transition of bacteria into the persister or VBNC state.
- How was thermal stability and stability to pH changes assessed? Was the activity of Q7 tested on all strains (evaluating MIC for each), or only on one strain? This information is missing.
- There is no information about the error bars or p-values in the results presented in Figure 1. The figure legend also lacks an explanation of antibiotic abbreviations.
- I suggest providing a legend explaining the colors of the "boxes" next to the table, rather than in the description, especially since the white boxes are visible in the table as empty spaces (which suggests the test was not performed). Additionally, color interpretation could be misleading. I perceive the box representing intermediate resistance as pink rather than brown.
- To assess the degree of bacterial eradication, the results of IP and ATP could be utilized. However, in this case, the results should be compared with the negative control (dead bacteria).
4. Why was the LN175 strain used for SEM, TEM, and transcriptome sequencing? Why were different experimental parameters used (Q7 at 0.25 x MIC, 4-hour incubation time) compared to the IP and NPN experiments (Q7 at 0.25–1 MIC, 30-minute incubation time)? Both of these experimental groups aimed to indicate the same effect of Q7, i.e., disrupting the integrity of the bacterial cell membrane. This requires clarification.
- In the figure legend for Figure 2, it is stated that the incubation time with Q7 in the IP and NPN assays was 30 minutes, while in subsection 2.11, there is mention of 30 minutes followed by 1 hour (lines 155-156).
- There is an error in the graph legend for Figure 2A regarding ATP. On the 'y' axis, it should read "luminescence intensity" instead of "fluorescence intensity."
5. What is the frequency of cell wall perforation events after Q7 treatment, as assessed by SEM? In Figure 2E, one cell with a visible perforation is shown. Statistical data should be provided, indicating the frequency of perforations (cells in which perforation was observed) relative to the total number of cells in the sample, in order to exclude random events. The same applies to the TEM studies and the identification of cell wall crumpling.
6. Were bacterial CFUs in the blood measured, or was blood culture performed, to investigate the presence of bacteria in the blood?
- There is a lack of consistency in the reported measurable quantities. In the Materials and Methods, the gavage dose is given in CFU/kg body weight of the rat (line 90), whereas in the Results, the gavage dose is mentioned in mg/ml (line 246). What does this value refer to?
- In the description of the experiment (analysis of IL-18, IL-6, and TNF-alpha levels), the term "medicated groups (D and E; line 154)" is used. However, neither in the methods section (subsection 2.3) nor in Figure 4 are these symbols associated with the appropriate experimental groups (they refer to subsequent sections in Figure 4).
- The figure legend for Figure 4 is incomplete. There is no information on the type of tissue presented in section A. The mention of 'H&E staining' for A-D is provided, where B-D are graphs, but there is no legend for graphs B-G. It would be helpful to clarify what the abbreviations CT and the symbols ++ and + next to Q7 represent.
7. The complete results from the transcriptomic analysis should be included in the manuscript as supplemental data.
8. The Discussion section contains many repetitions from the Results section. I suggest emphasizing the advantages of Q7 over other antibiotics or other AMPs in this part of the manuscript. The manuscript presents the results of studies for polymyxin E, which were neither analyzed in the Results nor in the Discussion.
9. It would also be beneficial to investigate the activity of Q7 against persistent bacteria and those in the VBNC state.
10. The scale bar should be indicated on the microscope images, as is done in the original images. Including the description "scale bar = 1 µm" under the images is unclear, especially when referring to images with different magnifications (e.g., Figure 2 panels E and F).
11. I suggest improving the graphical abstract to make it more understandable.
12. Style and Nomenclature:
- The names of bacterial species should be written in italics. Please ensure that when a species is first mentioned in the manuscript, its full name is used (Escherichia coli). Subsequent references to the species should be abbreviated (E. coli).
- Gene names should be written in lowercase and italics (e.g., mlaC, https://biocyc.org/gene?orgid=ECOLI&id=G7659).
- Please use the correct names for reagents (line 119: NaCl and HCl).
- Please follow the correct citation format according to the author guidelines (References must be numbered in order of appearance in the text).
- Keywords – please arrange them in alphabetical order.
- Line 122: repetition.
- Please standardize the style throughout the manuscript (e.g., p-value should be written in lowercase), and use lowercase for measurement units (e.g., mg/ml).
Author Response
To Reviewer #2:
Dear Reviewer, Thank you very much for your valuable comments. Your Suggestions have helped us to improve the deficiencies in the manuscript and increase the rigor and readability of the article. We have completed the revision and have responded to your comments point by point as follows:
Majors:
Comments 1:[Valuable information regarding E. coli, currently found in the 'Discussion' section (lines 277-293), should be moved to the introduction to better introduce the topic of pathogenicity.]
Response 1:[Thank you very much for your helpful comments.The discussion regarding the importance of combating E. coli pathogenicity has been moved to the introduction section.]
Comments 2:[I recommend expanding the introductory information on AMPs (lines 45-46), e.g., by providing a definition of AMPs, describing their general common mechanisms of action, and highlighting their broad range of activity. It would also be valuable to mention that the mechanism of action of peptides makes them active against both metabolically active and dormant bacteria (including persisters and VBNC) in contrast to conventional antibiotics, which target specific molecular mechanisms requiring bacterial activity and do not act on dormant bacteria]
Response 2:[Thank you very much for your valuable comments.The advantages of antimicrobial peptides over antibiotics, along with their action characteristics, are emphasized in the new introduction section.]
Comments 3:[ I suggest removing lines 53-55. The content of this part is not coherent with the rest of the paragraph. It could be integrated into the paragraph discussing the modification introduced to D-CONGA (to explain why this modification was made]
Response 3:[Thank you very much for your helpful guidance. This section has been deleted]
Comments 4:[ Please revise lines 57-63, as it is unclear, especially lines 57-58. Two peptides are described, but one name is used, which is confusing. As I understand, D-CONGA was initially selected (Starr et al, 2020), and then it was modified by Ghimire et al.,(2023), and named D-CONGA-Q7]
Response 4:[Thank you for your thorough review and we apologize for our carelessness.The description of the Q7 has been changed in the new introduction]
Comments 5:[Is the use of the name Q7 in the manuscript simply an abbreviation for D-CONGA-Q7? This should be clarified.]
Response 5:[Thank you for your careful inspection,We have changed D-CONGA-Q7 to Q7 in the article to avoid potential confusion.]
Comments 6:[ Lines 31-33: Lack of references. Lines 33-34: Lack of references.]
Response 6:[Thank you for your careful inspection,References have been added in lines 31-34]
Comments 7:[Information about NPN and PI should be moved from subsection 2.1 to subsection 2.11.]
Response 7:[Thank you very much for your careful review,NPN, PI related information has been moved to the appropriate location]
Comments 8:[ I would suggest expanding the information on the strains used in the study (a description of these strains or relevant references), or explaining why these specific strains were chosen for the experiments. This is particularly important for the selection of the K88 strain for mice infections (the information in lines 293-294 should be moved to the beginning of the 'Results' section or included in the description of strains in the 'Materials and Methods' section) and LN175 for other experiments.]
Response 8:[Thank you for your valuable feedback on our paper.Information has been updated in the Material Methods, Results, and Discussion in the newly submitted manuscript]
Comments 9:[ There is no information on how the MIC of the antibiotics was measured, even though line 18 states: "we evaluated the MIC values of common antibiotics...".]
Response 9:[Thank you very much for your valuable comments.Information about MIC and MBC determination is described in 2.5 and 2.6.]
Comments 10:[There is no information on which strain was tested with PI and NPN or what concentrations of Q7 were added to the samples in subsection 2.11. This becomes clear only in figure 2A.]
Response 10:[Thank you very much for your careful review,In the new manuscript, the corresponding concentrations of the Q7 have been made up in the appropriate places]
Comments 11:[ There is no description of how the small intestine tissue samples were collected, fixed, and stained, nor how the microscopic observations were conducted.]
Response 11:[Thank you very much for your careful review.In the new manuscript, detailed experimental steps on HE staining have been added at 2.5]
Comments 12:[ Subsection 2.4 does not specify which antibodies were used (company, origin).]
Response 12:[Thank you very much for your helpful comments.We determined the Elisa indicators using ELISA kits (Jianglai biology, Shanghai, China), which contain the corresponding antibodies.]
Comments 13:[ There is no mention of the specific equipment used for the experiments. For example, while a scanning electron microscope is mentioned, its exact model and the name or parameters of the camera used to capture the images are not provided.]
Response 13:[Thank you very much for your kind attention,The type of transmission electron microscope as well as the scanning electron microscope has been indicated in the appropriate place.]
Comments 14:[ The manuscript presents MIC results for Q7 and other antibiotics. MIC determines the concentration that inhibits bacterial growth, but it is not synonymous with the Minimum Bactericidal Concentration (MBC). The absence of growth does not imply that the bacterial cells are dead; they may be in a dormant state (persisters) or in a VBNC state. In the Materials and Methods section, it is stated that MBC was evaluated (subsection 2.6), but these results are not presented or analyzed. In the 'Results' section, there is no mention of MBC, but it is stated that "Q7 effectively eradicated multidrug-resistant E. coli with MIC values below 4 µg/ml" (lines 186-187). This statement is also mentioned in the manuscript title. MIC does not equate to bacterial eradication. MBC results should be provided, demonstrating that Q7 does not induce the transition of bacteria into the persister or VBNC state.]
Response 14:[Thank you for your careful inspection,The MIC of Q7 in Figure 1A has been changed to MBC and the MBC of Q7 has been emphasized in subsequent analyses to demonstrate that Q7 can completely eradicate bacteria]
Comments 15:[How was thermal stability and stability to pH changes assessed? Was the activity of Q7 tested on all strains (evaluating MIC for each), or only on one strain? This information is missing.]
Response 15:[Thank you very much for your kind attention,The antimicrobial activity of Q7 after various temperatures as well as ph treatments was determined using LN175, J27ab-2, and K88 and has been described in a new manuscript]
Comments 16:[There is no information about the error bars or p-values in the results presented in Figure 1. The figure legend also lacks an explanation of antibiotic abbreviations.]
Response 16:[Thank you very much for your careful review,The full names of the antibiotics in Figure 1A have been updated and the error lines in Figures 1B and C have been filled in]
Comments 17:[I suggest providing a legend explaining the colors of the "boxes" next to the table, rather than in the description, especially since the white boxes are visible in the table as empty spaces (which suggests the test was not performed). Additionally, color interpretation could be misleading. I perceive the box representing intermediate resistance as pink rather than brown.]
Response 17:[Thank you very much for your kind attention,The abbreviation of the antibiotic in Figure 1A has been changed to the full name, and a legend has been added to Figure 1A for clarity, and the color information has been corrected in the annotation]
Comments 18:[ To assess the degree of bacterial eradication, the results of IP and ATP could be utilized. However, in this case, the results should be compared with the negative control (dead bacteria).]
Response 18:[Thank you for your kind comments.Based on your suggestion, we have replaced the MIC data for evaluating bacteria with MBC data .]
Comments 19:[ Why was the LN175 strain used for SEM, TEM, and transcriptome sequencing? Why were different experimental parameters used (Q7 at 0.25 x MIC, 4-hour incubation time) compared to the IP and NPN experiments (Q7 at 0.25–1 MIC, 30-minute incubation time)? Both of these experimental groups aimed to indicate the same effect of Q7, i.e., disrupting the integrity of the bacterial cell membrane. This requires clarification.]
Response 19:[Thank you very much for your kind attention,Because LN175 was the most antibiotic-resistant strain in our pre-MIC experiments, and it has both mcr-1 and NDM resistance genes, and it is resistant to both polymyxin and meropenem, which are the two antibiotics of the last line of defense, and is a super-resistant strain, we chose it to be the subject of sem,tem and transcriptome sequencing. In order to investigate the mechanism of action of Q7 on E. coli, we finally chose a 4h treatment time for transcriptome sequencing and electron microscopy observation, as the significance of transcriptome and electron microscopy changes increases with the increase of treatment time. For NPN and PI, we chose a 1h treatment time for membrane permeability experiments because the permeability changes could be significantly observed in 1h.]
Comments 20:[in the figure legend for Figure 2, it is stated that the incubation time with Q7 in the IP and NPN assays was 30 minutes, while in subsection 2.11, there is mention of 30 minutes followed by 1 hour (lines 155-156)]
Response 20:[Thank you for your thorough review and we apologize for our carelessness.Inconsistencies in legends and material methods have been corrected in a new manuscript]
Comments 21:[ There is an error in the graph legend for Figure 2A regarding ATP. On the 'y' axis, it should read "luminescence intensity" instead of "fluorescence intensity."]
Response 21:[Thank you for your thorough review and we apologize for our carelessness.The title of the ATP y-axis has been corrected in the new manuscript]
Comments 22:[What is the frequency of cell wall perforation events after Q7 treatment, as assessed by SEM? In Figure 2E, one cell with a visible perforation is shown. Statistical data should be provided, indicating the frequency of perforations (cells in which perforation was observed) relative to the total number of cells in the sample, in order to exclude random events. The same applies to the TEM studies and the identification of cell wall crumpling.]
Response 22:[Thank you for your kind comments.We re-looked at SEM as well as TEM and found that perforation of bacterial cell membranes under SEM was indeed a random phenomenon, and we didn't find more perforated bacteria, so we counted the percentage of bacterial cell membranes that were wrinkled under SEM and the percentage of perforated contents that oozed out in TEM and tabulated them to put in a new manuscript]
Comments 23:[Were bacterial CFUs in the blood measured, or was blood culture performed, to investigate the presence of bacteria in the blood?]
Response 23:[Thank you for your kind comments.When we did the experiment of Q7 for the mouse inflammation model, the purpose at that time was to prove its anti-inflammatory effect, and we did not count the number of bacteria in the blood, which may be a defect of our experiment, and we will consider in this direction in the future related experiments.]
Comments 24:[ There is a lack of consistency in the reported measurable quantities. In the Materials and Methods, the gavage dose is given in CFU/kg body weight of the rat (line 90), whereas in the Results, the gavage dose is mentioned in mg/ml (line 246). What does this value refer to?]
Response 24:[Thank you very much for your kind attention,We have changed it accordingly in the new manuscript]
Comments 25:[ In the description of the experiment (analysis of IL-18, IL-6, and TNF-alpha levels), the term "medicated groups (D and E; line 154)" is used. However, neither in the methods section (subsection 2.3) nor in Figure 4 are these symbols associated with the appropriate experimental groups (they refer to subsequent sections in Figure 4).]
Response 25:[Thank you very much for your kind attention,We have changed the description of the groups in the new manuscript and added a detailed description in the legend of Figure 4]
Comments 26:[ he figure legend for Figure 4 is incomplete. There is no information on the type of tissue presented in section A. The mention of 'H&E staining' for A-D is provided, where B-D are graphs, but there is no legend for graphs B-G. It would be helpful to clarify what the abbreviations CT and the symbols ++ and + next to Q7 represent.]
Response 26:[Thank you for your thorough review and we apologize for our carelessness.We have corrected the information about the organization of the small intestine in the new manuscript and indicated the meaning of “+”, “++”, and “-”.]
Comments 27:[The complete results from the transcriptomic analysis should be included in the manuscript as supplemental data.]
Response 27:[Thank you very much for your kind attention,We will upload data from transcriptome sequencing in the Supplementary Material of the new manuscript.]
Comments 28:[The Discussion section contains many repetitions from the Results section. I suggest emphasizing the advantages of Q7 over other antibiotics or other AMPs in this part of the manuscript. The manuscript presents the results of studies for polymyxin E, which were neither analyzed in the Results nor in the Discussion.]
Response 28:[Thank you very much for your kind attention,In our new manuscript, we have strengthened the emphasis on the advantages of Q7 and its advantages over other antibiotics, as well as the fact that it has a therapeutic efficacy comparable to that of polymyxin E in the treatment of small bowel inflammation models]
Comments 29:[ It would also be beneficial to investigate the activity of Q7 against persistent bacteria and those in the VBNC state.]
Response 29:[Thank you very much for your kind attention,In our new manuscript, we replaced the evaluation of Q7's bactericidal effect with MBC]
Comments 30:[ The scale bar should be indicated on the microscope images, as is done in the original images. Including the description "scale bar = 1 µm" under the images is unclear, especially when referring to images with different magnifications (e.g., Figure 2 panels E and F).]
Response 30:[Thank you for your thorough review and we apologize for our carelessness.In our new manuscript, we have added the corresponding scales to the different scales of the electron microscope maps]
Comments 31:[ I suggest improving the graphical abstract to make it more understandable.]
Response 31:[Thank you very much for your kind attention,We have added HE staining to the new legend as well as a short view of Elisa to facilitate understanding of the]
Comments 32:[ The names of bacterial species should be written in italics. Please ensure that when a species is first mentioned in the manuscript, its full name is used (Escherichia coli). Subsequent references to the species should be abbreviated (E. coli).Gene names should be written in lowercase and italics (e.g., mlaC, https://biocyc.org/gene?orgid=ECOLI&id=G7659).]
Response 32:[Thank you very much for your kind attention,We have italicized the corresponding terms in the new manuscript, and have abbreviated the multiple occurrences throughout]
Comments 33:[ Please use the correct names for reagents (line 119: NaCl and HCl).]
Response 33:[Thank you for your thorough review and we apologize for our carelessness.We have written corrections for HCl, NaOH, and NaCl in the new manuscripts]
Comments 34:[Please follow the correct citation format according to the author guidelines (References must be numbered in order of appearance in the text). Keywords – please arrange them in alphabetical order. Line 122: repetition.Please standardize the style throughout the manuscript (e.g., p-value should be written in lowercase), and use lowercase for measurement units (e.g., mg/ml).]
Response 34:[Thank you for your thorough review and we apologize for our carelessness.We have corrected the order of literature citations in the new manuscript. And the keywords have been sorted alphabetically. And the units of measurement as well as the p-values have been lower-cased]
Reviewer 3 Report
Comments and Suggestions for Authors
The study is dedicated to elucidate the molecular mechanisms of activity of the antimicrobial peptide Q7 against drug-resistant E. coli strains. Despite a lot of methods involved, I did not find any novelty in this work. The ability of antimicrobial peptides (AMPs) to disrupt bacterial membranes has been known for decades, and there are dozens of models of this process (reviewed, for example, in 10.1021/acs.chemrev.8b00520). For each exact AMP, a structure-function relationship must be understood, i.e. the peptide structure must be analyzed with respect to the membrane composition. The same applies to the interaction of AMPs with the membranes of cellular organelles in the analysis of the inflammatory response. However, this information is completely missing in the present study. Therefore, I suggest the authors to analyze the structure of Q7 peptide together with the composition of bacterial membranes and to prove their findings in model systems such as the commonly used GUV models.
Reviewer 4 Report
Comments and Suggestions for Authors
Dear authors, greetings!
The manuscript "The antimicrobial peptide Q7 eradicates drug-resistant E. coli by disrupting bacterial cell membranes" investigates the potential of this peptide to eliminate this Gram-negative species, the mechanism through which it occurs and also the protective effect the drug exerts in small intestine.
Special attention is required from authors regarding italics and punctuation. Escherichia coli, E. coli, in vivo, in vitro are terms that need to be in italics. There are various lines in which punctuation and/or the use of capital letters is incorrect (for example, 20, 57, 76, 90, 119…).
Abbreviations also need attention; it is necessary to define the meaning of NPN, PI, MHB, MH and DCFH-DA in the first time the terms are mentioned.
When it comes to introduction, estimation on the future problem to be faced regarding resistance to antibiotics should be included to highlight the importance of this study (for example, number of deaths to be caused).
Regarding research design, the study lacks western blot to prove alteration in the expression pattern of products from genes MlaC, YcfL and any other relevant to the study.
In material and methods, it is necessary to mention the origin of Q7: was it donated, bought …? Subsection 2.2 needs to be written in past tense. In subsection 2.3. the number of the approved study needs to be added. In subsection 2.7, the text from line 120 to 123 is confusing and needs to be re-written. It is also necessary to explain why the concentrations tested were chosen. Subsection 2.10 needs to present details on procedures performed and pieces of equipment used to allow readers to repeat the process if they want to. The process regarding HE staining needs to be described.
When it comes to results, Figure 1A lacks presentation of standard deviation regarding results from Q7’,s MIC. Figures 1B-1E lack standard deviation bars associated with replications. Figures 2E-2F lack scale bar. A Table containing the 64 genes upregulated and the 24 ones downregulated needs to be presented.
In discussion it is necessary to remove number 1 from line 261 and add the correct reference. This section needs to be improved comparing results to others associated with antibacterial peptides present in literature. The other genes with an altered expression pattern also need to receive attention.
After western blot analysis, the conclusion needs to be reviewed to reflect all data presented.
Author Response
To Reviewer #4:
Dear Reviewer, Thank you very much for your valuable comments. Your Suggestions have helped us to improve the deficiencies in the manuscript and increase the rigor and readability of the article. We have completed the revision and have responded to your comments point by point as follows:
Comments 1:[Special attention is required from authors regarding italics and punctuation. Escherichia coli, E. coli, in vivo, in vitro are terms that need to be in italics. There are various lines in which punctuation and/or the use of capital letters is incorrect (for example, 20, 57, 76, 90, 119…).]
Response 1:[Thank you for your thorough review and we apologize for our carelessness.In the new manuscript, the relevant terms have been italicized and the case has been changed]
Comments 2:[Abbreviations also need attention; it is necessary to define the meaning of NPN, PI, MHB, MH and DCFH-DA in the first time the terms are mentioned.]
Response 2:[Thank you for your thorough review and we apologize for our carelessness.The full name of the corresponding term has been included in its first occurrence.]
Comments 3:[When it comes to introduction, estimation on the future problem to be faced regarding resistance to antibiotics should be included to highlight the importance of this study (for example, number of deaths to be caused).]
Response 3:[Thank you very much for your kind attention,We have emphasized the importance of antibiotic resistance in introduction]
Comments 4:[Regarding research design, the study lacks western blot to prove alteration in the expression pattern of products from genes MlaC, YcfL and any other relevant to the study.]
Response 4:[Thank you very much for your kind attention,We also found this problem during the experiment, but due to the lack of mlaC and ycfL related antibodies, this experiment could not be carried out, however, our lab is working on the knockout of the gene mlaC in order to prove the role of this gene, and we will seriously consider your suggestion in the future experiments.]
Comments 5:[In material and methods, it is necessary to mention the origin of Q7: was it donated, bought …? Subsection 2.2 needs to be written in past tense. In subsection 2.3. the number of the approved study needs to be added. In subsection 2.7, the text from line 120 to 123 is confusing and needs to be re-written. It is also necessary to explain why the concentrations tested were chosen. Subsection 2.10 needs to present details on procedures performed and pieces of equipment used to allow readers to repeat the process if they want to. The process regarding HE staining needs to be described.]
Response 5:[Thank you very much for your kind attention,In the new manuscript, information on the source of Q7 has been provided, 2.2 has been modified to be written in the past tense, subsection 2.7 has been modified accordingly for ease of comprehension, 2.10 has been supplemented with information on the equipment, and a new subsection has been added to write a detailed step-by-step procedure for HE staining]
Comments 6:[When it comes to results, Figure 1A lacks presentation of standard deviation regarding results from Q7’,s MIC. Figures 1B-1E lack standard deviation bars associated with replications. Figures 2E-2F lack scale bar. A Table containing the 64 genes upregulated and the 24 ones downregulated needs to be presented.]
Response 6:[Thank you for your thorough review and we apologize for our carelessness.In the new manuscript, Figure 1A has been revised and the error lines in Figure 1 B-E have been filled in, and a table of the transcriptome sequencing data will be included with the submission of the new manuscript]
Comments 7:[In discussion it is necessary to remove number 1 from line 261 and add the correct reference. This section needs to be improved comparing results to others associated with antibacterial peptides present in literature. The other genes with an altered expression pattern also need to receive attention.]
Response 7:[Thank you for your thorough review and we apologize for our carelessness.In the new manuscript, the cited information has been corrected, and other genetic changes are analyzed in detail in a future article]